# *YAP* silencing by *RB1* mutation is essential for small-cell lung cancer metastasis

Zhengming Wu ⊕[1], Junhui Su[2], Fu-long Li[3], Tao Chen ⊕[4], Jaimie Mayner[5], Adam Engler[5], Shenghong Ma[6], Qingquan Li ⊕[2] ✉ & Kun-Liang Guan[1,7] ✉

Small cell lung cancer (SCLC) is highly lethal due to its prevalent metastasis. Most SCLCs have inactivating mutations in *TP53* and *RB1*. We find that loss of *YAP* expression is key for SCLC cells to acquire rapid ameboid migration and high metastatic potential. YAP functions through its target genes *CCN1/CCN2* to inhibit SCLC ameboid migration. *RB1* mutation contributes to *YAP* transcriptional silencing via E2F7, which recruits the RCOR co-repressor complex to *YAP* promoter. We discover that benzamide family HDAC inhibitors stimulate *YAP* expression by inhibiting the RCOR-HDAC complex, thereby suppressing SCLC metastasis and improving survival in a mouse model. Our study unveils the molecular and cellular basis underlying SCLC's high metastatic potential, the previously unrecognized role of YAP in suppressing ameboid migration and tumor metastasis, and the mechanism of *YAP* transcription regulation involving E2F7, RCOR, and Sin3 HDAC. This study reveals a therapeutic potential of benzamides for SCLC treatment.

Lung cancer is the leading cause of cancer death in the US and the world. Based on histology phenotypes, lung cancers can be classified to two major types: non-small cell lung carcinoma (NSCLC) and small cell lung carcinoma (SCLC). The majority of lung cancers are NSCLC, while about 10-15% of lung cancers are SCLC[1]. However, SCLC is the most aggressive lung cancer with a poor prognosis, a five-year survival of less than 5%[2–4]. Metastasis is a top concern for SCLC patients. Nearly 70% SCLC patients have distant metastases at diagnosis, commonly found in the lymph nodes, brain, liver, and bones. Despite extensive research, the reason that SCLC has such a remarkable metastatic proclivity is still unknown.

Different from other lung cancers, a striking genetic feature of SCLC is the almost uniform inactivation of both *TP53* and *RB1* tumor suppressors. SCLC is believed to be originated from pulmonary neuroendocrine cells (PNEC)[5]. In genetic mouse models, deletion of *RB1* and *P53* in PNECs leads to development of lung cancer similar to human SCLC. Most SCLC display many of the neuroendocrine markers and are known as NE-SCLC. A small fraction of SCLC does not express

neuroendocrine markers and is classified as non-NE SCLC. An interesting feature of SCLC is that SCLC cells, unlike most solid tumors, do not adhere to normal culture plates and grow in suspension in vitro.

There are limited treatment options, generally including cytotoxic chemotherapies. SCLC patients often show good response to chemotherapy initially, but most patients will rapidly develop resistance in a few months and eventually succumbed to the metastatic cancer. The only major improvement in the standard therapy for SCLC over the past 30 years was the recent approval of immune checkpoint inhibitors, which have extended survival by several months, but the response rate is low[3,4]. Prophylactic cranial irradiation reduces the appearance of brain metastases; however, this treatment modality does not consistently improve disease-free or overall survival. Therefore, identifying new approaches to limit metastases is critical for SCLC treatment.

YAP is a transcription co-activator downstream of the Hippo pathway. YAP is generally considered as an oncoprotein to promote cell proliferation and inhibit apoptosis. Surprisingly, *YAP* expression is

[1]Department of Pharmacology and Moores Cancer Center, University of California, San Diego, La Jolla, CA 92093, USA. [2]Department of Pharmacology, School of Pharmacy, Fudan University, Shanghai 201203, China. [3]College of Life Sciences, Zhejiang University, Hangzhou, Zhejiang 310058, China. [4]Endoscopy Center, Department of Gastroenterology, Shanghai East Hospital, School of Medicine, Tongji University, Shanghai 200120, China. [5]Department of Bioengineering, University of California, San Diego, La Jolla, CA 92093, USA. [6]Wellcome Sanger Institute, Cambridge CB10 1RQ, UK. [7]Present address: School of Life Sciences, Westlake University, Hangzhou 310030, China. ✉e-mail: 061101040@fudan.edu.cn; guankunliang@westlake.edu.cn

strongly repressed in SCLC. Here, we found that *YAP* transcription repression is the key molecular event for SCLC cells to gain ameboid fast migration and high metastatic potential. YAP silencing is caused by *RB1* mutation. *RB1* loss acts via E2F7 elevation, which recruits the RCOR repressors to silence *YAP* transcription. Furthermore, benzamide family histone deacetylase (HDAC) inhibitors, such as entinostat, induce *YAP* expression by inhibiting the RCOR-HDAC (also known as CoREST) complex. Interestingly, *YAP* transcription is repressed by the RCOR-HDAC complex, but stimulated by the SIN3-HDAC complex. Entinostat, by virtue of RCOR-HDAC inhibition, impedes SCLC ameboid migration in vitro and metastasis in mouse models in a manner dependent on YAP induction. Our study reveals previously unrecognized roles of RB1 and YAP in SCLC metastasis, the mechanism of E2F7, RCOR, and Sin3 in *YAP* transcriptional regulation, and therapeutic potential of benzamide for SCLC treatment.

## Results

### YAP silencing in SCLC is obligatory for tumor metastasis

YAP is a transcription co-activator in the Hippo signaling pathway and is often elevated in cancer. Through pan-cancer bioinformatics analysis, we found that *YAP* expression is strongly downregulated in *RB1* mutant SCLC cell lines, but not non-small cell lung cancer cell lines (NSCLC) (Supplementary Fig. S1a, b). We performed lung cancer tumor samples with a YAP antibody and confirmed *YAP* downregulation in SCLC tumors but not in lung adenocarcinoma or squamous cancer samples (Fig. 1a). Notably, non-NE cells in SCLC samples are YAP positive (Supplementary Fig. S1c).

YAP normally promotes cell growth and is generally considered to exert oncogenic function[6]. The dramatic downregulation of *YAP* in SCLC is rather intriguing. To investigate the functional significance of *YAP* downregulation in SCLC, we established stable H209 cell pools with doxycycline (Dox) inducible *YAP* (Fig. 1b). *YAP* induction had no obvious effect on H209 cell growth in vitro or tumor growth in vivo (Fig. 1b). *YAP* induction had no obvious effect on cell death (Supplementary Fig. S1f). Further, *YAP* induction did not affect the expression of neuroendocrine SCLC markers *ASCL1* and *SYP* (Supplementary Fig. S1g). In mice transplanted with control H209 cells, metastases were readily detected in contralateral lung, mediastinal lymph node, axillary lymph node, liver, chest wall, brain, bone, and diaphragm within six weeks after transplantation (Fig. 1c, Supplementary Fig. S1h). In contrast, mice that received the inducible-*YAP* H209 cells showed a dramatic reduction in tumor metastases, although the primary tumor growth was not significantly affected. We performed similar *YAP*-inducible experiments with two additional SCLC cell lines H69 and H209 (Supplementary Fig. S1d). Again, *YAP* induction had little effect on cell growth in vitro or primary tumor growth, but strongly blocked tumor metastasis (Supplementary Fig. S1e, i, j). These results demonstrate that YAP inhibits metastasis in SCLC, and its transcriptional repression is obligatory for the high metastatic potential of SCLC.

To determine the effect of YAP during different phases of metastasis, we administered Dox at different times following mouse xenograft. One day of Dox treatment was sufficient to induce *YAP* expression in primary tumors (Supplementary Fig. S1k). Dox administration on day 2 post xenograft, a time point that distal micrometastasis had not been formed, blocked liver metastasis (Fig. 1d) and improved survival (Fig. 1e). At day 35, a time point at which distal micrometastasis had already been established, Dox treatment did not reduce metastasis load (Fig. 1d). Therefore, *YAP* expression appears to suppress early events in metastasis but not the growth of established metastatic SCLC foci.

To further characterize the function of YAP in metastasis, we expressed EGFP in control (EGFP⁺/YAP⁻) and mScarlet in *YAP* inducible (mSCARLET⁺/YAP⁺) SCLC cell lines H209 and H526. Equal amounts of the EGFP or mSCARLET-labeled cells were mixed and then xenografted into mice. Dox treatment robustly and selectively reduced circulating mSCARLET⁺/YAP⁺ H209 or H526 cells as determined by flow cytometry for EGFP or mSCARLET positive cells (Fig. 1f). Quantitative PCR for human *GAPDH* confirmed that *YAP* induction dramatically reduced circulating human tumor cells (CTCs) (Fig. 1g). We also examined the composition of mSCARLET⁺/YAP⁺ and EGFP⁺/YAP⁻ cells in the primary and metastatic tumors. The primary tumors were composed of both mSCARLET⁺/YAP⁺ and EGFP⁺/YAP⁻ cells (Fig. 1h). In contrast, no mSCARLET⁺/YAP⁺ cells were present in liver metastases. These observations support a model that YAP inhibits the initial stages of SCLC metastasis by reducing circulating tumor cells prior to distal colonization.

### YAP inhibits SCLC ameboid migration

Unlike most solid tumor cells, SCLC cells exhibit non-adhesive growth in vitro, thus they lack ECM dependent mesenchymal-like migration. However, previous studies have shown that under confinement conditions, which may more mimic in vivo condition, SCLC cells display ameboid migration. We performed SCLC migration assays using polydimethylsiloxane (PDMS) with either 3 μm diameter beads or fabricated 3 μm pillars (Supplementary Fig. S2a, b)[7,8]. Cell death was not observed under our confinement setting (Supplementary Fig. S2c). SCLC cells, including H209, H69, and H526, displayed ameboid morphology and ameboid migration with an average speed of 6 μm/min (Fig. 2a, Supplementary Fig. S2d, e, Supplementary Movie 1). This is much faster than mesenchymal-like migration that is typically slower than 1 μm/min[9]. We also observed ameboid-like fast migration in tumors by imaging SCLC tumors in xenografted mice (Supplementary Movies 2–3). Our data indicate that the high-speed ameboid-like migration may be the cytological basis for the remarkable metastatic proclivity of SCLC.

To further understand the role of YAP silencing in SCLC metastasis, we explored whether YAP regulates SCLC migration. In the control SCLC cells, a high fraction of cells showed polarized cell morphology, characterized by a stable pear-like shape and large spherical protrusion front (Supplementary Fig. S2f) and fast migration velocity (Fig. 2a, Supplementary Fig. S2d, e, Supplementary Movie 1)[10]. *YAP* induction strongly inhibited the ameboid morphology and fast migration in SCLC cells (Supplementary Movie 4). These observations suggest that YAP may inhibit SCLC ameboid migration to suppress tumor metastasis although further study is required to unambiguously demonstrate a causal relationship.

We examined the effect of *YAP* transcriptional activity on ameboid morphology and migration. Expression of either WT *YAP* or the constitutively active S127A *YAP* mutant, but not the inactive *YAP* S127D mutant, reduced ameboid cell population and migration in H209, H69 or H526 cells (Fig. 2b, Supplementary Fig. S2g, h). The *YAP* S94A mutant, which is transcriptionally inactive due to defective TEAD binding, failed to inhibit ameboid migration, indicating that YAP requires its transcription activity to inhibit ameboid migration. Together, we propose that *YAP* silencing represents a key molecular event contributing to the high metastatic potential of SCLC.

### YAP acts through CCN1/2 to inhibit SCLC migration

To understand the mechanism of YAP in regulating SCLC cell migration, we performed RNA-seq and ATAC-seq of parental and *YAP* re-expressing H526 cells. GO analysis revealed enrichment in processes related to cytoskeleton organization and Rho GTPase in *YAP*-re-expressing H526 cells (Fig. 2c), consistent with YAP's role in SCLC migration. *YAP* expression did not affect actin levels in H209 or H526 cells (Supplementary Fig. S2i). We used Lifeact-TagGFP to monitor actin dynamics in H209, H69, and H526 cells (Supplementary Fig. S2j; Supplementary Movies 5, 6)[11]. Under confinement, a stable polar cortical actin density gradient was observed in the parental cells. However, F-actin was non-polar and dim in *YAP* re-expressing cells (Fig. 2d; Supplementary Fig. S2k). Moreover, *YAP* re-expression also reduced

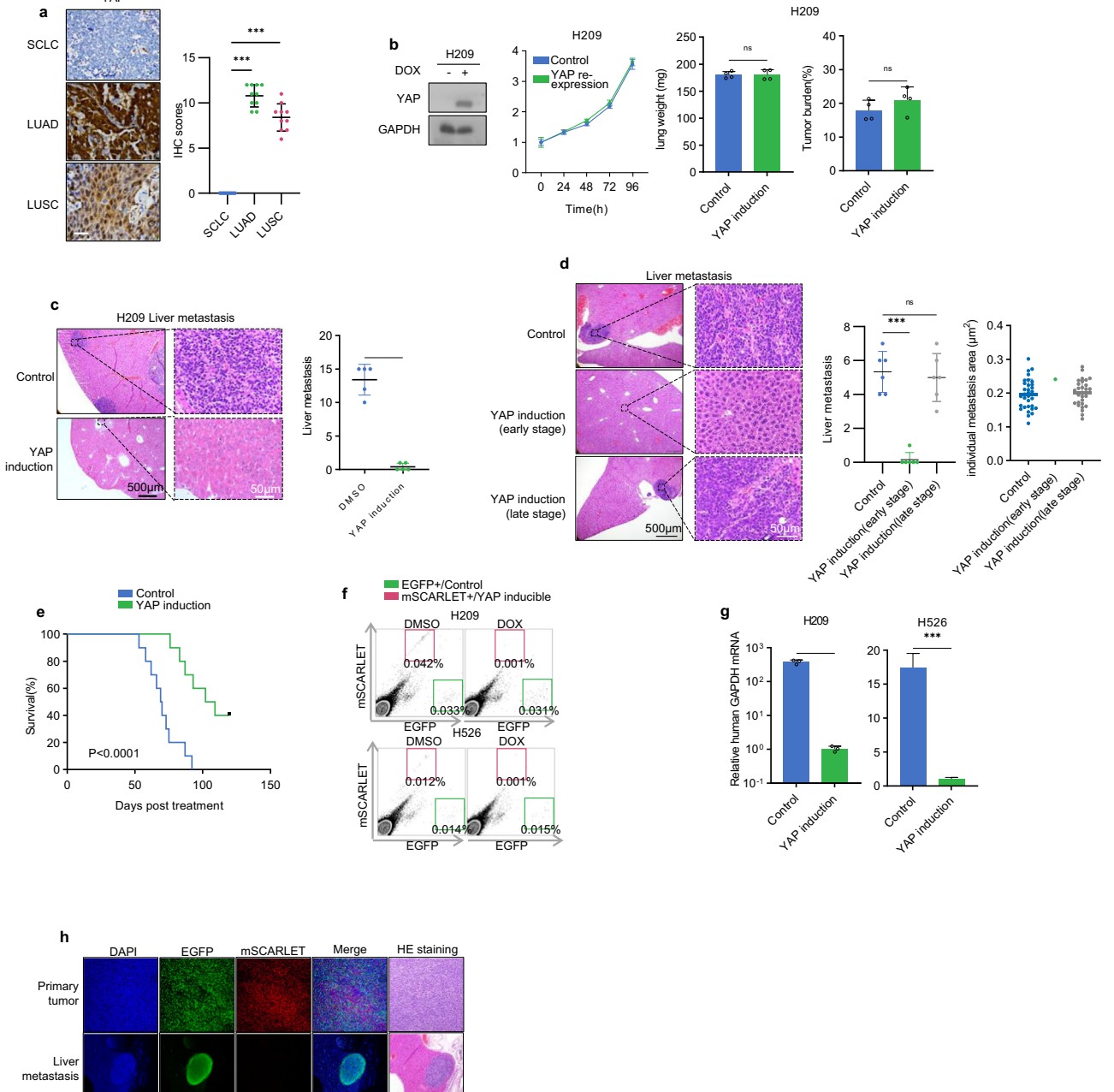

**Fig. 1 | *YAP* is silenced in SCLC and re-expression inhibits tumor metastasis. a** *YAP* is expressed in lung adenocarcinoma (LUAD) and squamous carcinoma (LUSC) but not in SCLC tumors. YAP protein in tumors was detected by a specific antibody, and the dot plot on the right shows the IHC quantification of 10 samples for each cancer subtype. **b** *YAP* induction does not affect the growth of H209 in vivo and tumor formation in vitro. DOX denotes the presence of 100 ng/mL Dox in culture. Viable cells were determined by the CCK8 assay. **c** *YAP* induction inhibits H209 liver metastasis. Liver from vehicle and Dox-treated mice were stained with HE (left panels). Quantification of liver metastasis foci per mouse is shown in the right panel. **d** *YAP* induction at the early but not late stage inhibits liver metastasis. Liver from control and early/late Dox-treated mice were stained with HE (left panels). Quantification of metastatic sites per mouse is shown in the right panel. **e** The survival curve of nude mice after orthotopic injection of control or *YAP* inducible H209 cells. **f** YAP decreases CTCs. Equal amounts of EGFP-labeled control and mSCARLET-labeled *YAP* inducible cells were injected into mouse lungs. Blood cells were collected and analyzed for EGFP (*x* axis) and mSCALET (*y* axis) to quantify CTCs (green or red boxes). **g** *YAP* re-expression inhibits CTCs. Relative CTCs were determined by human GAPDH mRNA normalized against mouse GAPDH mRNA. **h** EGFP⁺/YAP⁻ cells but not mSCARLET⁺/YAP⁺ form liver metastasis. Mice were orthotopically injected with a mixture of both cells. Source data are provided as a Source data file.

F-actin polymerization in primary tumors of H209 and H526 xenografted mice (Fig. 2e; Supplementary Fig. S2l).

*CCN1/2* (also known as Cyr61/CTGF) were the top upregulated genes in *YAP*-induced H209, H69 and H526 cells (Supplementary Fig. S2m). They are secreted extracellular matrix (ECM) proteins and can impact cell adhesion, invasion, and migration[12–14]. We tested whether CCN1/2 might play a role in ameboid migration. We found that addition of recombinant CCN1 or CCN2 blocked F-actin polarization and ameboid migration under confinement (Fig. 2d, f; Supplementary Fig. S2k, n). We further assessed the importance of CCN1/2 in mediating the effect of YAP on SCLC cell migration. We generated *CCN1/2* double knockout cells and then inducibly expressed YAP. *CCN1/2* dKO

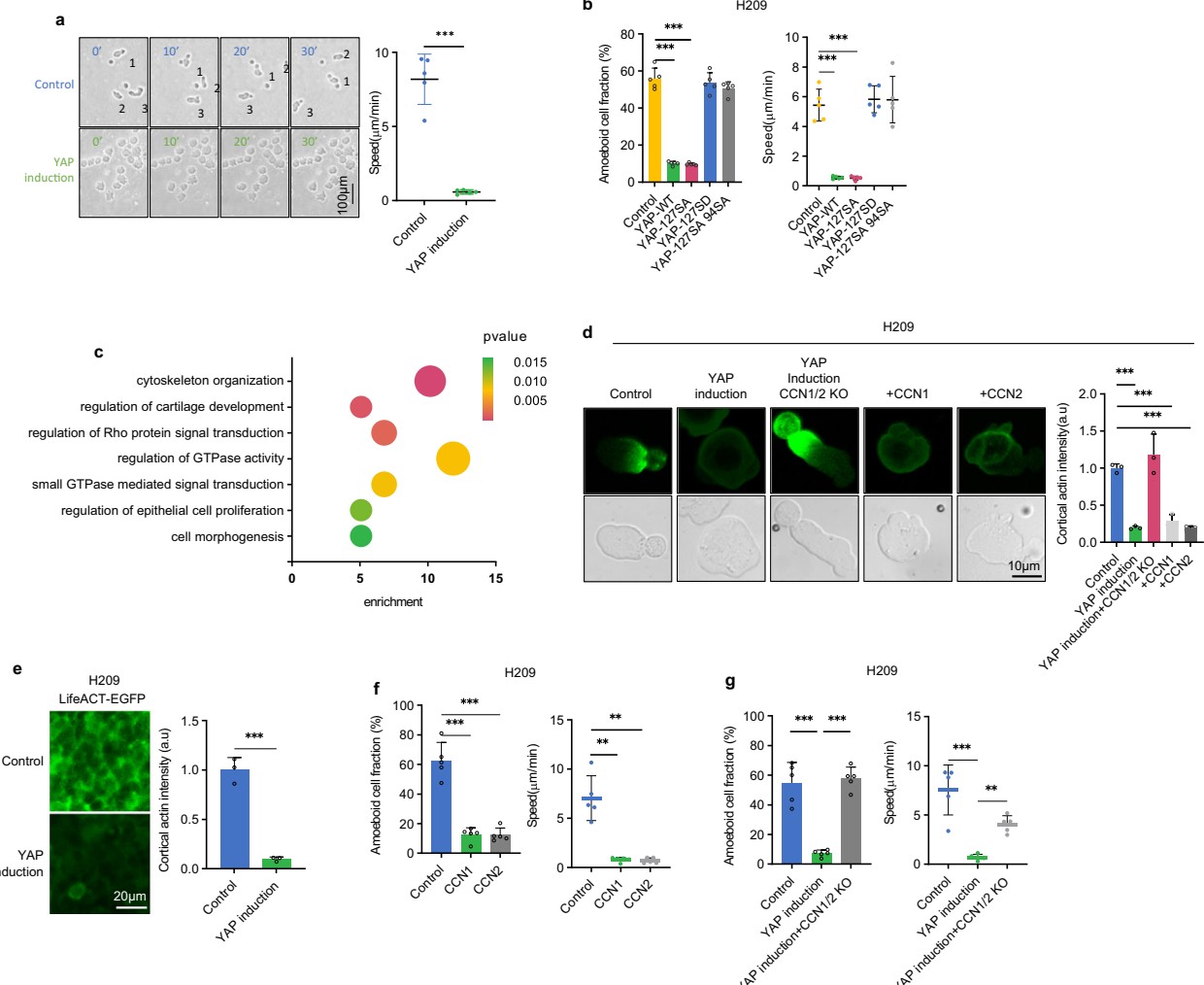

**Fig. 2 | YAP inhibits SCLC ameboid migration through CCN1/2. a** *YAP* expression inhibits the fast ameboid-like migration of H209 cells under confinement. Ameboid migration speed were quantified and shown on the right. **b** TEAD-dependent YAP co-transcriptional activity is required to suppress H209 ameboid migration. **c** *YAP* expression alters genes in the cytoskeleton and Rho GTPase pathways. GO pathway enrichment scatterplot shows the enriched DEG pathways in control and *YAP* re-expressed H526 cells. Dot size is proportional to the number of DEGs. **d** The effect of YAP and CCN1/2 on F-actin polarization. Fluorescence images of Lifeact-TagGFP are shown for H209 cells under confinement. **e** *YAP* re-expression decreases F-actin intensity in H209 xenografted tumors. Quantification is shown in the right panel. **f** Purified CCN1 or CCN2 protein inhibits ameboid migration in H209 cells. **g** CCN1/2 are required for YAP to inhibit H209 cells ameboid migration. Source data are provided as a Source data file.

blocked the effect of *YAP* expression on F-actin polarization and ameboid migration (Fig. 2d, g; Supplementary Fig. S2k, o). Altogether, our results suggest that CCN1/2 are key downstream effectors of YAP and may act as an autocrine or paracrine to inhibit SCLC ameboid migration.

## E2F7 and RCOR mediate the effect of *RB1* loss on *YAP* downregulation

*YAP* expression is strongly downregulated in *RB1* mutant SCLC cell lines, but not in *RB1* wild type SCLC or NSCLC (Supplementary Fig. S1a). Notably, *RB1* mutation does not affect *YAP* expression in NSCLC. To test whether RB1 and TP53 regulate *YAP* expression, we stably expressed *RB1* or *TP53* in SCLC lines H209, H69, and H526, which have mutations in both *TP53* and *RB1*. Re-expression of *RB1*, but not *TP53*, increased *YAP* and YAP target genes *CCN1/2* (Supplementary Fig. S3a–d), indicating that loss of *RB1* is responsible for *YAP* repression. It should be noted that *RB1* re-expression moderately reduced cell growth while *P53* re-expression strongly inhibited SCLC growth. By RNA sequencing, we examined the global expression profiles of genes regulated by YAP or *RB1* re-expression in H526 cells. Most YAP upregulated genes (239 out

of the 279) were also upregulated by *RB1* re-expression, consistent with *YAP* induction by RB1 re-expression. As expected, RB1 affected more genes than YAP because RB1 modulates other transcription factors, such as E2F1 (Fig. 3a). These results highlight a key role of *RB1* mutation in suppressing *YAP* expression and activity in SCLC.

RB1 generally exerts transcriptional control indirectly through interacting with partner proteins, such as the E2F family transcription factors[15]. We observed that *E2F7* was upregulated in *RB1* mutated SCLC cell lines but had no correlation with *RB1* mutations in other tumor types, such as breast cancer (Supplementary Fig. S3e–g). Unlike E2F1-5, which have RB1-binding domains, E2F6-8 do not have RB1 binding domains and function as transcription repressors[16–18]. We investigated the function of E2F7 in YAP regulation in SCLC. Deletion of *E2F7*, but not *E2F6* or *E2F8*, increased *YAP* mRNA and protein levels in H209, H69 and H526 cells (Fig. 3b; Supplementary Fig. S3h). Double deletion of *E2F7* and *E2F8* induced stronger *YAP* expression, indicating a key role of E2F7/8 in *YAP* silencing. Next, we examined the relationship between RB1 and E2F7 in modulating *YAP* expression. Re-expression of *RB1* reduced E2F7 mRNA and protein with a concomitant induction of *YAP* in H209, H69, and H526 cells (Fig. 3c, d; Supplementary Fig. S3i).

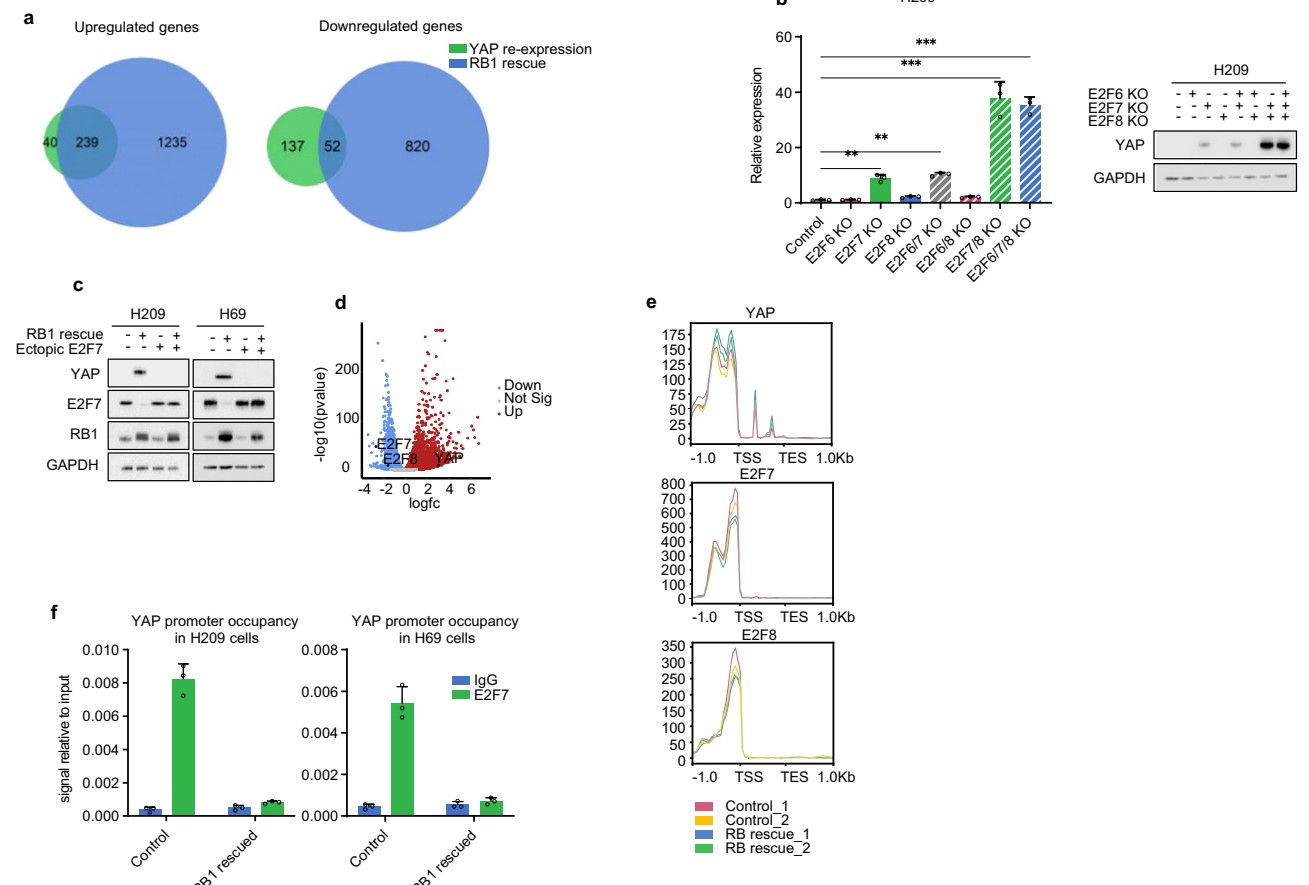

**Fig. 3 | RB1 loss results in YAP downregulation. a** Overlap of genes upregulated or downregulated by *YAP* re-expression or *RB1* rescue in H526 cells. Only differentially regulated genes (>2-fold and *p* < 0.05 compared to the control H526 cells) were included in the analysis. **b** *E2F7/8* KO induces *YAP* mRNA and protein in H209 cells. **c** Ectopic *E2F7* expression blocks *YAP* induction by RB1 in H209 and H69 cells. **d** Volcano plot showing the upregulation and downregulation of *YAP* and *E2F7/8* in H526 cells, respectively. **e** Effect of RB1 on chromatin accessibility in the promoters of *YAP*, *E2F7*, and *E2F8*. Control and *RB1* rescue H526 cells were compared. TSS and TES denote transcription start and end site, respectively. All genes have been normalized to a 1 kb length. **f** *RB1* re-expression abolishes E2F7 binding on the *YAP* promoter. Chromatin immunoprecipitation (ChIP) and RT-qPCR show E2F7 occupancy on the *YAP* promoter in control and *RB1* rescued H209 and H69 cells. Source data are provided as a Source data file.

Importantly, ectopic *E2F7* expression blocked *YAP* induction by *RB1* re-expression (Fig. 3c, d; Supplementary Fig. S3i). Together, our data suggest that E2F7 acts downstream of and mediates the effect of RB1 on *YAP* transcriptional regulation.

We performed assay for transposase-accessible chromatin with sequencing (ATAC-seq) and observed that *RB1* re-expression caused a moderate increase and decrease in chromatin accessibility in *YAP* and *E2F7/8* genes, respectively (Fig. 3e). To test whether E2F7 might directly repress *YAP*, we performed E2F7 chromatin immunoprecipitation (ChIP). E2F7 was found on the *YAP* promoter in H209, H69, and H526 cells and *RB1* re-expression decreased E2F7 occupancy on the *YAP* promoter (Fig. 3f; Supplementary Fig. S3j). Based on the above observations, we propose that RB1 loss allows *E2F7* expression to repress *YAP* transcription.

Little is known regarding the mechanism of E2F7 in transcription repression. To gain molecular insight into E2F7-mediated *YAP* repression, we searched for E2F7 interacting proteins. Flag-E2F7 was stably expressed in H526 cells and purified using anti-FLAG magnetic beads. Mass spectrometry analysis showed that E2F7 copurified with all components of the RCOR-HDAC1/2-KDM1A complex (Fig. 4a, Supplementary Table 1). Reciprocal immunoprecipitation and mass spectrometry with RCOR1 also identified E2F7 as an RCOR1 interacting protein along with other known RCOR1 complex proteins (Supplementary Fig. S4a, Supplementary Table 2). By co-immunoprecipitation, we

confirmed that RCOR1 interacted with E2F7, but not E2F1, in H209, H69, and H526 cells (Fig. 4b, c; Supplementary Fig. S4b, c).

To determine the function of RCOR in *YAP* regulation, we deleted *RCOR1/2/3* in SCLC cell lines. We found that knockout of *RCOR1/2/3* induced *YAP* mRNA and protein in H69, H209, and H526 (Fig. 4d; Supplementary Fig. S4d). The RCOR complex contains histone deacetylase and demethylase, and represses gene expression by altering histone modifications. Knockout of either *E2F7/8* or *RCOR1/2/3* elevated the activating histone marker H3 lysine 4 trimethylation (H3K4me3) at the *YAP* promoter (Fig. 4e). These results support a model that in *RB1* mutant SCLCs, the elevated E2F7 recruits RCOR complex to the *YAP* promoter to induce repressive histone modifications and transcriptional silencing.

## RB1 mutation promotes SCLC ameboid migration via YAP repression

Since *YAP* expression is regulated by RB1, we investigated the function of RB1 in SCLC migration. Ectopic *RB1* expression inhibited ameboid migration in H209, H69, and H526 cells (Fig. 4f; Supplementary Fig. S4e), revealing a critical role of *RB1* mutation in promoting SCLC migration. Importantly, *YAP* knockout completely blocked the inhibitory effect of *RB1* re-expression on ameboid migration in these cells (Fig. 4f; Supplementary Fig. S4e). Consistent with their role in *YAP* regulation, knockout of *E2F7/8* or *RCOR1/2/3* abolished ameboid

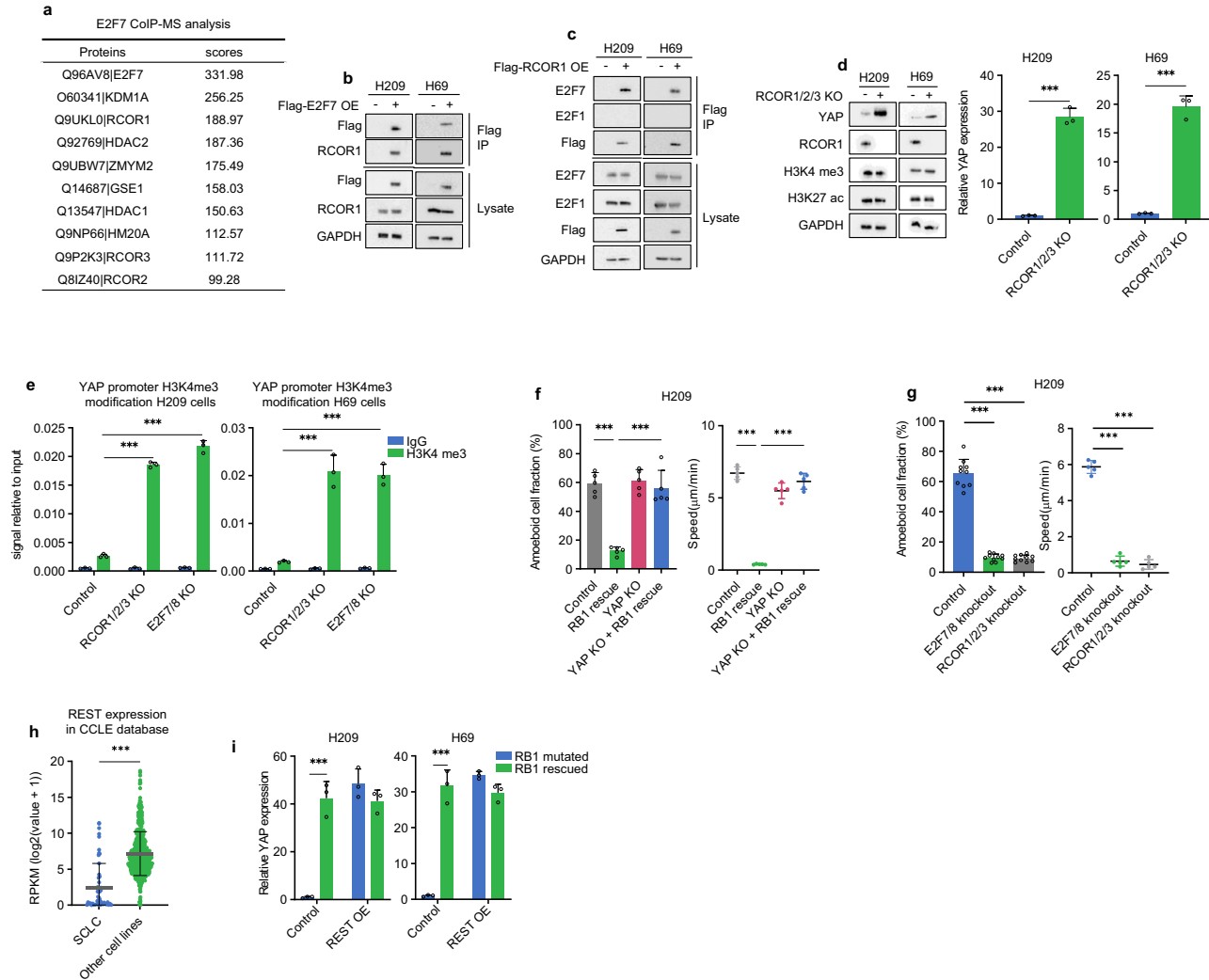

**Fig. 4 | E2F7 and RCOR mediate the effect of *RB1* loss on *YAP* downregulation.**
**a** Mass spectrometry showing the Flag-E2F7 interacting proteins in H526 cells. The background of proteins recovered from the Flag vector control expressing cells has been subtracted. The 10 top enriched proteins are listed. **b** E2F7 interacts with RCOR1. Co-immunoprecipitation (Co-IP) experiment confirmed the interaction between Flag-tagged E2F7 and endogenous RCOR1 in H209 and H69 cells. **c** RCOR1 interacts with E2F7 but not E2F1. Co-IP showed the interaction between Flag-RCOR1 and endogenous E2F family proteins in H209 and H69 cells. **d** *RCOR1/2/3* KO induces *YAP* mRNA and protein in H209 and H69 cells. **e** *RCOR1/2/3* and *E2F7/8* KO increases H3K4me3 *YAP* promoter of H209 and H69 cells. ChIP and RT-qPCR show *YAP* promoter H3K4 trimethylation. **f** YAP is required for RB1 to inhibit ameboid migration. Cell morphology (left) and migration (right) of control, *RB1* rescued, *YAP* KO, and *YAP* KO and *RB1* rescued H209 cells under confinement. **g** *E2F7/8 KO* or *RCOR1/2/3 KO* inhibits ameboid migration. Control, *E2F7/8* KO, and *RCOR1/2/3* KO H209 cells under confinement. **h** *REST* expression of SCLC cell lines and others in CCLE database. **i** Ectopic expression of *REST* in H209 and H69 cells induces *YAP* expression and abolishes the effect of RB. Source data are provided as a Source data file.

morphology in H209, H69, and H526 cells (Fig. 4g; Supplementary Fig. S4f). These data reveal a functional relationship that RB1 acts through *YAP* to inhibit ameboid migration, whereas *RB1* loss represses *YAP* through E2F7-RCOR to promote SCLC ameboid migration.

*YAP* regulation by RB is not a general phenomenon in all cell types, and is possibly limited to SCLC. We speculate that REST may play a key role in this cell-type specific effect as *REST* is expressed in virtually all cell types except neuronal and NE cells (Fig. 4h; Supplementary Fig. S4g). Previous studies have shown that REST recruits RCOR complex to repress gene expression in non-neural cells[19]. We observed that ectopic expression of *REST* in H209 and H69 cells increased *YAP* mRNA and importantly abolished the effect of *RB* re-expression on *YAP* (Fig. 4i). The above observations indicate that the absence of REST may allow E2F7 to bind with RCOR and repress *YAP* in SCLC.

Collectively, our results show that *RB1* inactivation promotes SCLC ameboid fast migration in a manner dependent on *YAP*

repression. YAP has little effect on cell growth per se, but acts as a tumor suppressor in SCLC by inhibiting cell migration and metastasis.

## Benzamide family HDAC inhibitors, but not TSA, induce *YAP* expression

Since the RCOR complex is known to repress gene expression by altering epigenetic modifications, we explored the effects of compounds known to inhibit histone and DNA modifications on *YAP* expression. Entinostat, a benzamide class I HDAC inhibitor[20], induced *YAP* and *CCN2* in H209, H69, and H526 cells (Fig. 5a, b; Supplementary Fig. S5a, b). Notably, the effect of entinostat on *YAP* expression was cell-type-specific since it did not induce *YAP* in HEK293T cells (Supplementary Fig. S5c). Surprisingly, trichostatin A (TSA), a more potent and broad spectrum HDAC inhibitor, did not induce *YAP*, but it did increase global H3K27 acetylation (Supplementary Fig. S5d, e)[21]. *HDAC1/2/3* knockout also did not induce *YAP* in H209 cells (Supplementary Fig. S5f). Interestingly, TSA treatment blocked entinostat-induced *YAP*

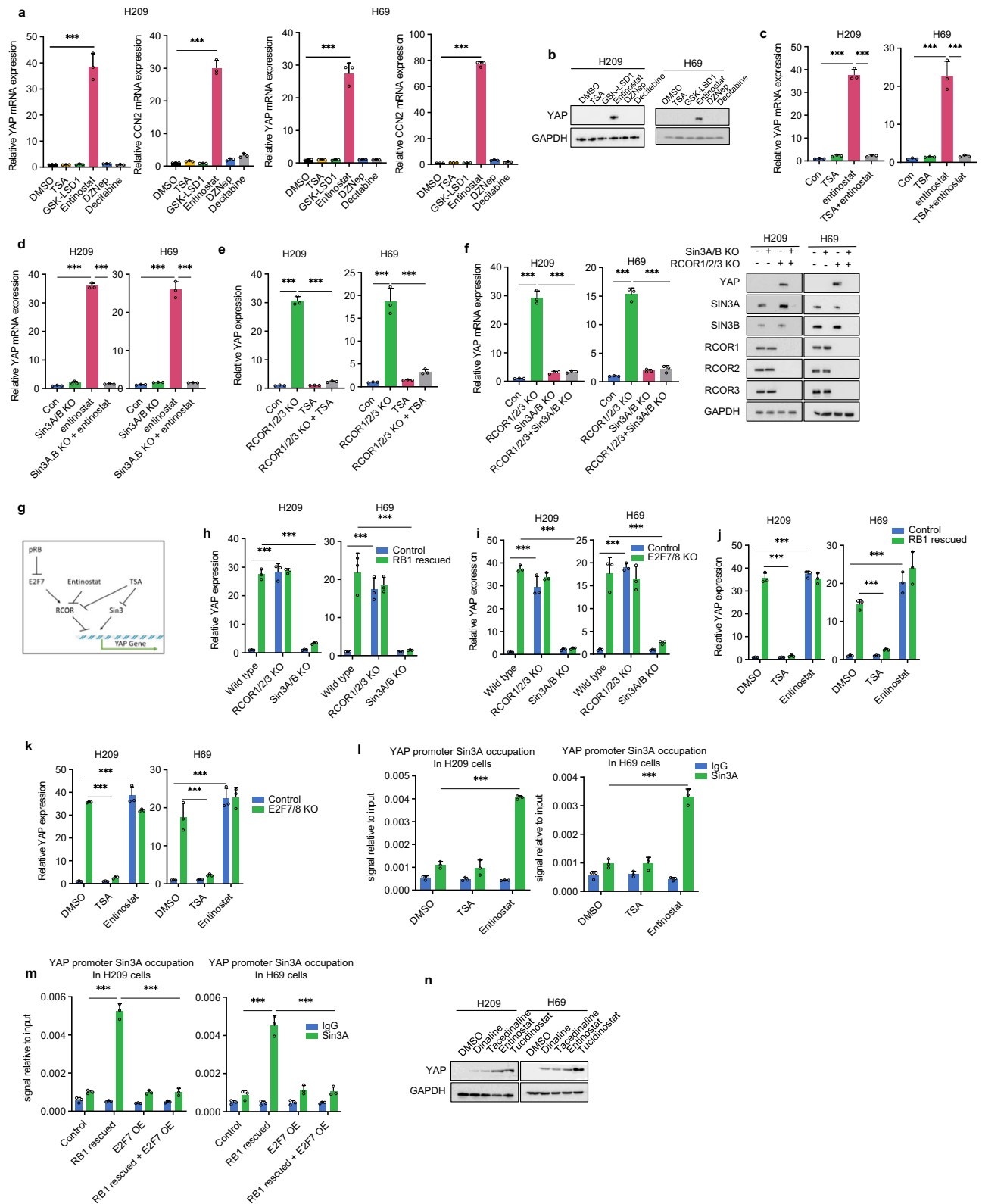

**YAP transcription is positively and negatively regulated by SIN3-HDAC and RCOR-HDAC, respectively**

HDACs form various protein complexes to confer distinct cellular functions[22]. Previous study by Bantscheff et al. showed that benzamide

expression (Fig. 5c; Supplementary Fig. S5g), indicating that different HDACs may have opposite effects on *YAP* expression and entinostat induces *YAP* possibly by selectively inhibiting an HDAC subgroup.

HDAC inhibitors were selectively towards RCOR, NURD, and NCOR complexes, but not the SIN3 complex, while TSA inhibited them all[23]. Since RCOR1/2/3 represses *YAP* expression, we hypothesized that entinostat induces *YAP* by inhibiting RCOR1/2/3-HDAC. TSA could not induce *YAP* because it inhibits more HDAC complexes, such as SIN3A/B-HDAC that may play a positive role in *YAP* expression. To test this hypothesis, we performed combinatory knockout experiments. *SIN3A/B* dKO had a minor effect on *YAP* expression in H209 and H69

**Fig. 5 | Benzamide family HDAC inhibitors induce *YAP* transcription via the SIN3A-HDAC complex. a** Entinostat induces *YAP* and *CCN2* expression. H209 and H69 cells were treated with 1 μM entinostat, 0.3 μM TSA, 1 μM GSK-LSD1, 1 μM DZNep, and 10 μM decitabine for 24 hours. **b** 1 μM entinostat treatment for 24 hours induces YAP protein in H209 and H69 cells. **c** TSA prevents entinostat from inducing *YAP* expression in H209 and H69 cells. **d** *SIN3A/B* KO prevents entinostat from inducing *YAP* expression in H209 and H69 cells. **e** TSA treatment blocks *YAP* induction by *RCOR1/2/3* KO in H209 and H69 cells. **f** *SIN3A/B* KO suppresses *YAP* mRNA and protein induction by *RCOR1/2/3* KO in H209 and H69 cells. **g** A proposed mechanism of *YAP* transcription regulation. RB1 acts through the E2F7-RCOR axis, while SIN3A/B is required for *YAP* expression in SCLC. **h** KO of *SIN3A/B* but not *RCOR1/2/3* blocks *YAP* induction by *RB1* re-expression in H209 and H69 cells. **i** *SIN3A/B* KO but not *RCOR1/2/3* KO blocks *YAP* induction by *E2F7/8* KO in H209 and H69 cells. **j** TSA suppresses *RB1* rescue-induced *YAP* expression in H209 and H69 cells. **k** TSA suppresses *E2F7/8* KO-induced *YAP* expression in H209 and H69 cells. **l** Entinostat but not TSA increases SIN3A occupancy on the *YAP* promoter. **m** *E2F7* expression blocks *RB1*-induced SIN3A occupancy on the *YAP* promoter. **n** Benzamide HDAC inhibitors induce *YAP* protein in H209 and H69 cells. Source data are provided as a Source data file.

cells. Importantly, *SIN3A/B* dKO blocked *YAP* induction by entinostat (Fig. 5d). Consistently, TSA blocked *YAP* induction by *RCOR1/2/3* tKO (Fig. 5e). Furthermore, *SIN3A/B* dKO blocked *YAP* induction by *RCOR1/2/3* tKO (Fig. 5f). These results support a model that SIN3A/B is required for *YAP* expression, whereas RCOR1/2/3 represses *YAP* expression (Fig. 5g).

Next, we examined the relationship between RB1 and HDACs in *YAP* expression in SCLC. *SIN3A/B* dKO blocked *YAP* induction by *RB1* expression (Fig. 5h). *SIN3A/B* dKO also blocked *YAP* induction by *E2F7/8* dKO (Fig. 5i). TSA attenuated *YAP* induction by *RB1* re-expression or *E2F7/8* dKO (Fig. 5j, k). Entinostat did not further induce *YAP* in *RB1* rescued or *E2F7/8* dKO cells (Fig. 5j, k). There is no additive effect on *YAP* induction by *RB1* expression, *E2F7/8* dKO, and entinostat, indicating they act in a linear pathway. These data further support our proposed model of *YAP* regulation in SCLC (Fig. 5g).

To further explore the mechanism of *YAP* regulation, we analyzed the ENCODE data and found that SIN3A is enriched around the *YAP* promoter in many cancer cell lines that are known to express YAP (Supplementary Fig. S5h). We performed chromatin immunoprecipitation and observed little basal SIN3A on the *YAP* promoter in H209 and H69 cells (Fig. 5l). However, SIN3A binding to the *YAP* promoter was increased by entinostat, but not TSA, suggesting that RCOR inhibits SIN3A binding to the *YAP* promoter. *RB1* re-expression also increased SIN3A binding to the *YAP* promoter, and this effect of RB1 was blocked by *E2F7* overexpression (Fig. 5m). Moreover, we observed RCOR1 binding to the *YAP* promoter under basal conditions in SCLC. This RCOR1 occupancy on the *YAP* promoter was attenuated by both entinostat and TSA (Supplementary Fig. S5i). *RB1* re-expression reduced RCOR1 binding on the *YAP* promoter, and this effect of RB1 was blocked by *E2F7* overexpression (Supplementary Fig. S5j). These data are consistent with the YAP regulation model in Fig. 5g.

All benzamide HDAC inhibitors stimulated *YAP* expression in H209, H69, and H526, with increasing potency from dinaline, tacedinaline, entinostat, to tucidinostat (Fig. 5n; Supplementary Fig. S5k, l). Altogether, our results reveal a mechanism of *YAP* regulation in SCLC by RB1 and entinostat. RB1 downregulates E2F7-RCOR, which antagonizes the positive effect of SIN3A/B, to induce *YAP* (Fig. 5g). Entinostat induces *YAP* expression by selectively inhibiting the RCOR-HDAC complex.

### Entinostat inhibits SCLC metastasis and extends survival

Consistent with *YAP* induction, we found that entinostat inhibited ameboid morphology and F-actin in H209, H69, and H526 cells (Fig. 6a, b; Supplementary Fig. S6a, b). Entinostat treatment had little effect on tumor growth at primary graft sites but strongly blocked liver metastasis in mice orthotopically inoculated with H209 or H526 cells (Fig. 6c; Supplementary Fig. S6c, d). Further characterizations showed that entinostat-induced *YAP* expression, diminished cortical F-actin in primary tumors, and reduced circulating tumor cells (CTCs) of both H209 and H526 cells (Fig. 6d–f; Supplementary Fig. S6e, f). Moreover, entinostat treatment significantly extended the survival of the H526 xenografted mice (Fig. 6g). These data indicate an exciting therapeutic potential of using entinostat for SCLC treatment.

We investigated the function of YAP in mediating the entinostat response using *YAP* KO H209, H69, and H526 cells. In contrast to the YAP wild type cells, entinostat treatment could neither reduce F-actin nor block ameboid cell migration in the *YAP* knockout cells in vitro, indicating an obligatory role of YAP in these entinostat-induced cellular responses (Fig. 6h, i, Supplementary Fig. S6g, h). These observations also indicate that the activities of entinostat on SCLC are unlikely due to general nonspecific effects. Consistently, entinostat failed to decrease F-actin, CTCs, and liver metastasis in mice inoculated with the *YAP* knockout cells (Fig. 6j–l, Supplementary Fig. S6i–k). Our data show that entinostat acts through *YAP* induction to inhibit SCLC metastasis.

## Discussion

Small cell lung cancer is notorious for its high metastasis while the underlying molecular mechanism remains elusive. This study identifies *YAP* silencing as a critical event in SCLC progression, enabling SCLC cells to acquire ameboid fast migration and high metastatic potential. *RB1* mutation is responsible for *YAP* silencing uniquely in SCLC. We propose that *YAP* silencing and ameboid migration as the underlying molecular and cellular basis for SCLC's high metastasis.

Different from classical mesenchymal migration, ameboid migration is much faster and independent of adhesion to ECM. Moreover, ameboid migration can pass through very narrow space[10]. YAP loss confers SCLC cells a strong cortical cytoskeleton which is required for ameboid migration, as ameboid migration mainly relies on cortical contraction rather than crawling[10]. It is worth noting that *YAP* is not expressed in hematopoietic cells, which also can perform ameboid migration. Therefore, YAP may have a general function to suppress adhesion independent ameboid migration.

Unlike other lung cancer types, virtually all SCLCs have mutations in *RB1* and *TP53*. RB1 is well-known as a tumor suppressor that blocks the cell cycle. This cell cycle inhibition model might be overly simplistic. Here, we have elucidated the molecular basis of *RB1* loss in *YAP* transcription silencing, mediated by E2F7 and RCOR-HDAC. We further showed that *YAP* transcription in SCLC is positively and negatively regulated by the SIN3-HDAC and RCOR-HDAC complexes, respectively. RB1 acts through inhibition of the RCOR-HDAC complex, enabling SIN3 to bind to the *YAP* promoter and stimulate *YAP* transcription, thereby suppressing SCLC metastasis. Our study reveals a key role of *RB* mutation in empowering the high metastatic potential of SCLC in a mechanism dependent on *YAP* transcriptional silencing.

YAP is a transcription co-activator and the main functional output of the Hippo tumor suppressor pathway, which is well-known for its physiological role in organ size control and tissue homeostasis[24–26]. *YAP* is generally considered to be oncogenic, as high YAP activity has been observed in many human cancer types[6,27]. For example, in mouse models, liver-specific *YAP* overexpression causes hepatomegaly, and sustained high YAP activity induces liver cancer[28,29]. Single-cell RNA sequencing databases have confirmed that PNECs have low *YAP* expression[30]. However, *YAP* expression is further silenced at transcriptional level dues to *RB* mutation in SCLC as revealed by this study. Our data shows that YAP has no direct effect on SCLC cell growth but rather acts as a tumor metastasis suppressor, potentially explaining

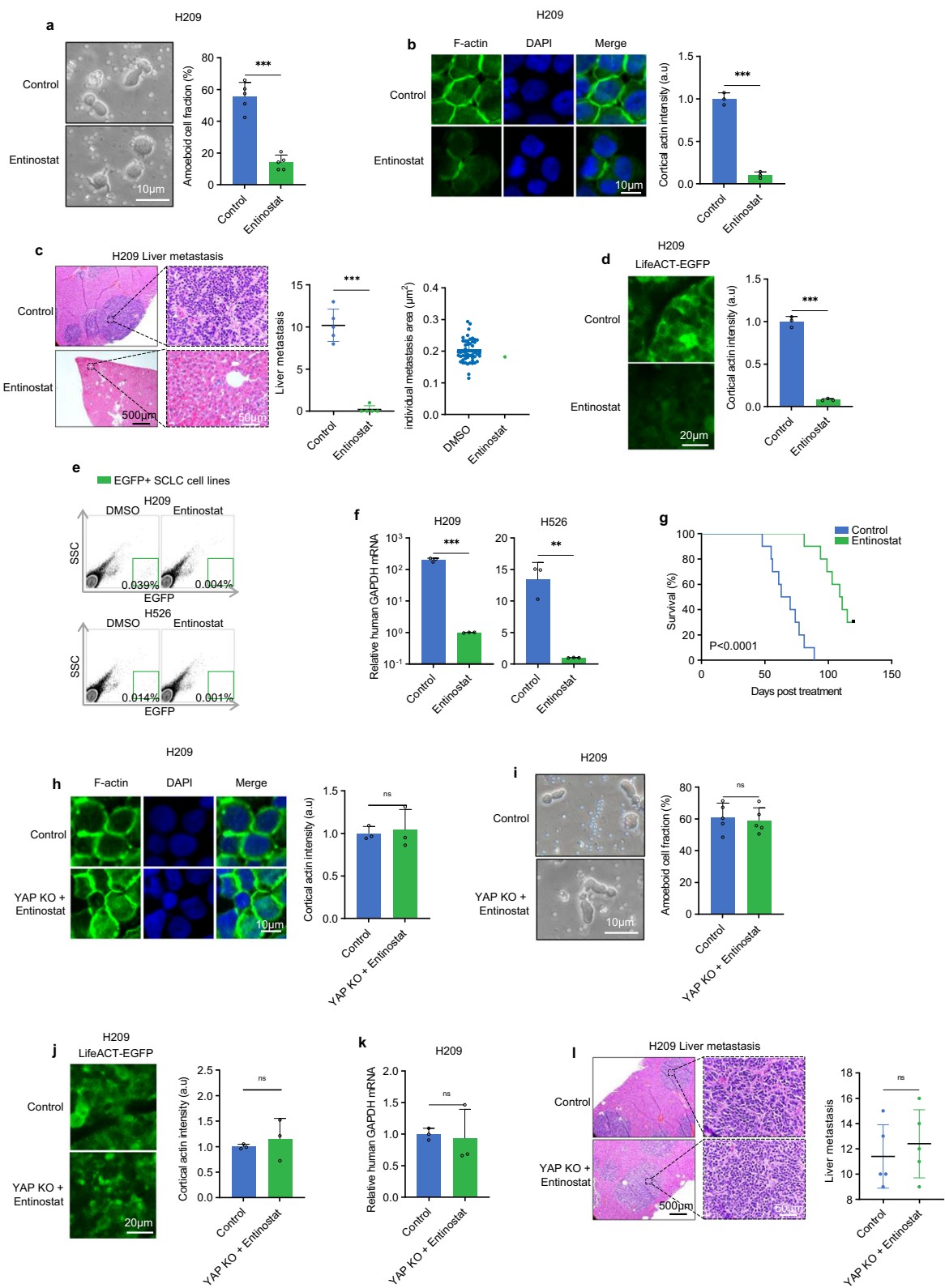

why *YAP* is silenced in SCLC. A tumor suppressor function of YAP in YAP-low cancers, including retinoblastoma and SCLC[31], has been suggested, and YAP signaling downregulation is also found to promote tumor progression in colorectal cancer[32]. Much study is needed to understand how YAP functions in these cancers. We further show that YAP target genes *CCN1/2* are required for YAP to suppress SCLC metastasis. Moreover, addition of recombinant CCN1 or CCN2 is sufficient to inhibit ameboid migration. Given that CCN1/2 are

extracellular proteins, we speculate that recombinant CCN1/2 proteins may inhibit SCLC metastasis.

SCLC cell lines can be classified into four subgroups: SCLC-A, SCLC-N, SCLC-P, and SCLC-Y[33]. Unlike the other three subtypes, SCLC-Y has *YAP* expression. The majority of SCLC cancers are NE like, including SCLC-A and SCLC-N, whereas SCLC-Y belongs to non-NE SCLC. We performed experiments in multiple SCLC cell lines: H209 and H69 that belong to the most common SCLC-A subtype, while H526

**Fig. 6 | Entinostat inhibits SCLC metastasis. a** Entinostat (1 μM, 24 hours) inhibits H209 cell ameboid migration under confinement. **b** Entinostat decreases H209 cell F-actin intensity in vitro. The fluorescence image and quantification of Lifeact-TagGFP are shown in the left and right panel, respectively. **c** Entinostat inhibits liver metastasis of orthotopically grafted H209 cells. HE staining shows liver metastasis. Quantification of metastasis sites per mouse is shown in the right panel. **d** Entinostat treatment decreases F-actin intensity in H209 xenografted tumors. Quantification is shown in the right panel. **e** Entinostat reduces CTCs in mice orthotopically inoculated with EGFP-labeled H209 or H526 cells. EGFP-positive CTCs in blood were detected by flow cytometry. **f** Entinostat reduces CTCs in mice orthotopically inoculated with either H209 or H526 cells. Experiments were similar to **d**, except CTCs were quantified by human *GAPDH* mRNA normalized against mouse *GAPDH* mRNA. **g** Entinostat prolongs survival of H209 xenografted mice. Nude mice were orthotopically inoculated with 5 × 10^6 H209 cells. Seven days later, mice were orally treated with a control vehicle or 12.3 mg/kg entinostat once daily, five days per week, until the terminal event. **h** Entinostat acts through YAP to decrease H209 cell F-actin intensity in vitro. The fluorescence image and quantification of Lifeact-TagGFP are shown in the left and right panel, respectively. **i** Entinostat acts through YAP to inhibit H209 cell ameboid cell migration. **j** Entinostat acts through YAP to inhibit F-actin in the H209 orthotopic model. Quantification is shown in the right panel. **k** *YAP* KO blocks the inhibitory effect of entinostat on CTCs in H209 cells. **l** YAP is required for entinostat to inhibit H209 liver metastasis. Source data are provided as a Source data file.

belongs to the SCLC-P subtype. Similar results were obtained in all three SCLC cell lines examined, suggesting a broad role of the RB1-YAP axis in modulating tumor metastasis and *YAP* transcription in SCLC.

YAP was reported to promote non-NE phenotypes in SCLCs in mice[34,35]. We observed that ectopic *YAP* expression induces genes in cell adhesion and focal adhesion in human SCLC cells. This can be interpreted that YAP induces non-NE cell like adhesion/motility characteristics. However, we also observed that *YAP* expression does not cause an overt NE to non-NE transition as NE markers are not downregulated. One possible explanation for the apparent discrepancy between the mouse data and our human cell data is the differences in genetic background between the in vivo mouse model and the in vitro human SCLC cells.

Entinostat and tucidinostat are clinical stage compounds as there are many ongoing clinical trials utilizing these compounds for cancer treatment (clinicaltrials.gov). Some results are available although most trials are ongoing. Entinostat is effective in relapsed or refractory Hodgkin's lymphoma whereas it is not effective in metastatic colorectal cancer or NSCLC (clinicaltrials.gov). The phase III clinical trial for advanced breast cancer concluded that combination of exemestane and entinostat did not improve survival in AI-resistant advanced HR-positive, HER2-negative breast cancer (clinicaltrials.gov). It is worth noting that both entinostat and tucidinostat are being tested for neuroendocrine tumors, but the results are yet to be reported. SCLC is one of the most aggressive cancers with limited treatment options. Our finding that entinostat inhibits SCLC metastasis has exciting therapeutic implication. Future studies are needed to investigate entinostat and tucidinostat for SCLC treatment.

## Methods
### Animal experiments
All mouse experiments were performed with procedures approved by the Institutional Animal Care and Use Committees (IACUCs) at the University of California-San Diego (UCSD; La Jolla, CA, USA) and Fudan University (Shanghai, China). Male nude (NU/J) mice were obtained from Jackson Laboratory (Bar Harbor, ME, USA). We used an optimized environment, including a 14/10- or 12/12-hour light/dark cycle, 18–23 °C temperatures with 40–60% humidity, and standard diet/water.

For lung orthotopic injection, 4-week-old male nude mice were anesthetized with sodium pentobarbital (50 mg/kg body weight) and placed in the right lateral decubitus position. A 5-mm skin incision overlying the left chest wall was made and the left lung was visualized through the pleura. A total of 1 × 10^6 NCI-H526/NCI-H209 cells (single-cell suspensions, greater than 90% viability) in 50 μg of growth factor-reduced Matrigel in 50 μL of Hank's balanced salt solution were injected into the left lungs of the mice through the pleura with a 30-gauge needle. After tumor cell injection, the wound was stapled and the mice were placed in the left lateral decubitus position and observed until fully recovered.

To administer vehicle or entinostat in vivo, we inserted a feeding needle into the mouse's mouth to a predetermined distance.

Entinostat or a vehicle were administered at 12.3 mg/kg once daily for five days per week for four weeks. For Dox induction in vivo, it was dissolved in water at 150 μg/ml. Treatment was continued until the control mice became moribund (50 days for the NCI-H526 or 40 days for the NCI-H209), at which point all mice were killed by $CO_2$ inhalation and assessed for peripheral blood, primary lung tumor, and distant metastasis.

### Human sections
Human lung cancer sections were obtained from patients in Huashan Hospital of Fudan University (Shanghai, China). All samples were obtained with informed consent from all subjects and in accordance with the ethical standards of the Institutional Review Board. Two or more pathologists independently reviewed the specimens, and a consensus diagnosis was reached based on the 2012 World Health Organization (WHO) diagnostic criteria.

### Cell culture
All cells were cultured at 37 °C and 5% CO2 in a humidified incubator. H69, H209, and H526 cells were maintained in a 10% fetal bovine serum (FBS) Roswell Park Memorial Institute (RPMI) 1640 culture medium (Gibco). 293 T cells were maintained in a 10% FBS Dulbecco's Modified Eagle Medium (DMEM) culture medium (Gibco).

### Regents
The regents used were: TSA (0.3 μM; Sigma-Aldrich, #T8552), entinostat (1 μM; Cayman Chemical, #13284), 3-deazaneplanocin A (DZNep; 1 μM; Cayman Chemical, #13828), rel-N-[(1 S,2 R)−2-phenylcyclopropyl]−4-piperidinamine (GSK-LSD1; 1 μM; Santa Cruz, #sc-490345), and 5-aza-2′-deoxycytidine (AZA; 10 μM; Sigma, #A3656), dinaline (1 μM; Abcam, #ab144561), tacedinaline (1 μM; MCE, #50934), tucidinostat (1 μM; MCE, #109015), Dox (100 nM; Cayman Chemical, #14422).

### Antibodies
The antibodies used were: ASCL1 (Abcam, #ab211327), INSM1 (Abcam, #ab170876), E2F7 (Abcam, ab245655), RB1 (Abcam, ab181616), SYP (Santa Cruz, #sc-17750), RCOR1 (Santa Cruz, #sc-376567), CCN1 (Santa Cruz, #sc-374129), CCN2 (Santa Cruz, #sc-365970), YAP/TAZ (Santa Cruz, #sc-101199), P53 (Santa Cruz, #sc-126), E2F1 (Santa Cruz, #sc-251), GAPDH (Santa Cruz, #sc-47724), LATS1 (Cell signaling, #3477), LATS2 (Cell signaling, #5888), p-LATS (Cell signaling, #8654), H3 (Cell signaling, #4499), H3K27ac (Cell signaling, #8173), H3K4me3 (Cell signaling, #9751), VIN (Cell signaling, #13901), Flag (Sigma, #1804),

### Cell confinement assay
Two methods were used to fabricate cell confinement chambers: (1) a chamber where confinement was fixed with borosilicate beads of a set diameter; (2) a chamber where PDMS pillar height sets the gap between its top and bottom, but the gap can be adjusted by compression. For method one[8], 2 mL polydimethylsiloxane (PDMS:crosslinker at 9:1) was mixed in each well of a 6-well plate. Precisely sized 3 μm diameter beads and cells were then loaded between the PDMS

and plate. For method two[7], molds were fabricated by standard photolithography, and large PDMS pillars were cast (PDMS:crosslinker at 35:1) into the custom-made mold. The smooth PDMS surface lacking pillars was bound to the plate lid by UV ozone treatment. Cells were cultured on the bottom plate with the top plate placed on top to confine the cells, with the relative degree of confinement and compression determined by pillar height and any additional weight applied to the top plate.

## Generation of KO and knock-in cells by CRISPR-Cas9

Gene KO and knock-in in this study were created with the CRISPR/Cas9 mediated gene editing system. The plasmids PX459 and lentiCRISPRv2 were obtained from Addgene (#62988 and #52961). The single guide RNA (sgRNA) sequences for KO were:

*RCOR1*: AGAAAAGCATGGGTACAACA;
*RCOR2*: TCACCCCATTCCCTGACGAG;
*RCOR3*: TAATGCCCGTTGGACCACAG;
*E2F6*: TTACCTACTTCTCTGGGAGC;
*E2F7*: ACAGACAGCAAGCGGAACCA;
*E2F8*: AAAACAGGTACACTTGGCAC;
*CCN1*: GGGCTGGTCCGGGACGGCTG;
*CCN2*: CCAGCTGCTTGGCGCAGACG;
*YAP*: GTGCACGATCTGATGCCCGG;

## Virus infection and ectopic gene expression

Cells stably ectopically expressing vector-based *RB1*, *TP53*, WT-*YAP*, *YAP*-S127A, *YAP*-S127D, *YAP*-S127A/S94A, Flag-*RCOR1*, Flag-*E2F7*, *CCN1*, and *CCN2* were created by lentivirus infection as described by Ma et al.

## Tumorigenesis assays

The lung orthoptic model was created by anesthetizing mice placed in the right lateral decubitus position. Next, $1\times10^6$ cells in 50 μL were mixed with 30% Matrigel and injected into the lung through a small skin incision in the left chest wall. Then, a metallic clip was used to close the skin. The mice were observed for 10 minutes until they fully recovered. Eight weeks later, mice were sacrificed by cervical dislocation, and their lung primary tumors and liver were harvested and fixed.

## Flag affinity purification

Control and H526 cells expressing Flag-tagged E2F7 or RCOR1 were lysed using a 1% NP40 buffer. The supernatant was harvested and incubated with an M2 Flag resin according to the manufacturer's instructions (Sigma-Aldrich, #A2220) overnight after 10 minutes of centrifugation at 15000 g. Then, the resin was washed with lysis buffer and eluted with 3XFlag peptide.

## Mass spectrometry

Mass-spectrometry analysis was performed on enriched proteins digested on beads by the UCSD Mass-Spectrometry Core. The data analysis was performed using the Byonic software (Protein Metrics).

## Circulating tumor cell assay with flow cytometry

After heparinizing mice, 500 μl of blood was collected into an anticoagulant tube to deplete red blood cells using a lysis solution. The remaining cells were resuspended in 1X HBSS containing 5% BSA. EGFP and mSCARLET-labeled tumor cells were analyzed using GFP and RFP channels, respectively in LSRFortessa X-20 by the UCSD Embryonic Core. The FlowJo software was used for final data processing.

## RNA-seq

Total RNA of each sample was extracted using RNeasy Mini Kit (Qiagen. Total RNA of each sample was quantified and qualified by Agilent 2100 Bioanalyzer (Agilent Technologies, Palo Alto, CA, USA), NanoDrop (Thermo Fisher Scientific Inc.) and 1% agrose gel. 1 μg total RNA with

RIN value above 6.5 was used for following library preparation. Next generation sequencing library preparations were constructed according to the manufacturer's protocol. The poly(A)mRNA isolation was performed using Poly(A) mRNA Magnetic Isolation Module. The mRNA fragmentation and priming was performed using First Strand Synthesis Reaction Buffer and Random Primers. First strand cDNA was synthesized using ProtoScript II Reverse Transcriptase and the second-strand cDNA was synthesized using Second Strand Synthesis Enzyme Mix. The purified double stranded cDNA by beads was then treated with End Prep Enzyme Mix to repair both ends and add a dA-tailing in one reaction, followed by a T-A ligation to add adaptors to both ends. Size selection of Adaptor ligated DNA was then performed using beads, and fragments of ~420 bp (with the approximate insert size of 300 bp) were recovered. Each sample was then amplified by PCR for 13 cycles using P5 and P7primers, with both primers carrying sequences which can anneal with flow cell to perform bridge PCR and P7 primer carrying a six-base index allowing for multiplexing. The PCR products were cleaned up using beads, and quantified by Qubit3.0 Fluorometer (Invitrogen, Carlsbad, CA, USA).

Read counts were determined using HT-Seq by counting the number of reads aligned by HISAT2 for each human transcript. We then used DEseq2 to determine differentially expressed genes between different treatment conditions.

## IHC score quantification

The scores represent the staining intensity and proportion of YAP positive cells. Specifically, IHC was scored by two independent pathologists. Based on the proportion of YAP positive stained-tumor cells which was assessed on the proportion score of 0–4: 0 (negative), 1 (1–25%), 2 (26–50%), 3 (51–75%), or 4 (76–100%) and the intensity score of 0–3: 0 (negative), 1 (weak), 2 (medium) or 3 (strong). The IHC score (ranging from 0 to 12) shown is the product of the proportion score times the intensity score.

## Analysis of survival data

All data were complete. Kaplan–Meier estimates of the survival function were plotted and used to compute median survival times.

## Statistical analyses

For non-survival experiments, two-tailed unpaired or one-way ANOVA analyses were used for comparison between groups in GraphPad Prism (GraphPad Software v9). All comparisons were two-sided unless specified otherwise. All analyzed P values are indicated for each comparison made within all figure panels. $P$ values of less than 0.05 were considered to indicate statistical significance.

## Supporting analyses

Immunoblot analysis, immunofluorescence, RNA isolation, real-time quantitative reverse transcription PCR (RT-qPCR), ATAC-seq, and their statistical analyses were performed as previously described by Ma et al.

## Reporting summary

Further information on research design is available in the Nature Portfolio Reporting Summary linked to this article.

## Data availability

The RNA-SEQ, ATAC-SEQ data generated in this study have been deposited in the SRA database under PRJNA752796 [https://www.ncbi.nlm.nih.gov/bioproject/PRJNA752796/]. Mass spectrometry data is available in the supplementary information file labeled supplementary table 1 and supplementary table 2. All other data supporting the findings of this study are available from the corresponding author upon reasonable request. Source data are provided with this paper.

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

## Acknowledgements

This work is supported by NIH grant CA268179 (K.L.G.). We thank Afshin Dowlati for sharing H526, H209 cell lines. We thank Xiaoyan Li, J. Matthew Franklin, and Ghassemian Majid for their technical assistance. We thank the support of the Office of Vice Chancellor for Health Sciences at UCSD. We thank Jean Y. Guan and J. Matthew Franklin for the critical reading of the manuscript.

## Author contributions

Z.W., Q.L., and K.-L.G. designed the study and analyzed the data. Z.W. and K.-L.G. wrote the manuscript. Z.W. performed most experiments with assistance from J.S., F.L., T.C., J.M., A.E., and S.M. All authors discussed the data and commented on the manuscript.

## Competing interests

K.-L.G. is a co-founder of and holds an equity interest in Vivace Therapeutics. All other authors declare no competing interests.
