## [Peer Review File · Nature Communications]

REVIEWERS' COMMENTS

Reviewer #1 (Remarks to the Author):

The revised manuscript has been thoughtfully amended in response to my initial feedback, and I am pleased to acknowledge the authors' dedicated efforts. The wealth of new data they have incorporated adds significant robustness to their conclusions, further accentuating the robustness of the study.

I do, however, have a minor suggestion for further clarification. The authors posit that Yap is downregulated in SCLC, and they present compelling evidence demonstrating its absence in this context. However, the relative framework for this downregulation remains somewhat ambiguous. Given that SCLC originate from pulmonary neuroendocrine cells, it would further bolster their argument if the authors could show that Yap is indeed expressed in these precursor cells. Providing such a comparison could lend even greater specificity and precision to their findings.

Reviewer #2 (Remarks to the Author):

The revised manuscript from Wu and colleagues is outstanding. The revision is highly responsive to the previous critiques and has now solidified the story in mechanistic detail. The authors have removed some distracting model systems from the previous submission which really improved the clarity of the findings. The data throughout the paper suggest the observations are biologically important and are robust across multiple cell lines. The insights gleaned are highly provocative as they tackle an important issue in SCLC research. The mechanism that links Rb-E2F7-HDAC-YAP to amoeboid cell migration and metastasis is novel and will be of great interest to the SCLC field.

minor points:

Line 155-156: The conclusion is too strong and should be softened. Yes both amoeboid migration and metastasis are inhibited by YAP expression, but these could be unrelated at this point. No experiments separate these phenomena functionally.

Line 290: Typo "F2F7" to "E2F7"

Line 290: Typo "selectively" to "selective"

Reviewer #1 (Remarks to the Author):

The revised manuscript has been thoughtfully amended in response to my initial feedback, and I am pleased to acknowledge the authors' dedicated efforts. The wealth of new data they have incorporated adds significant robustness to their conclusions, further accentuating the robustness of the study.

I do, however, have a minor suggestion for further clarification. The authors posit that Yap is downregulated in SCLC, and they present compelling evidence demonstrating its absence in this context. However, the relative framework for this downregulation remains somewhat ambiguous. Given that SCLC originate from pulmonary neuroendocrine cells, it would further bolster their argument if the authors could show that Yap is indeed expressed in these precursor cells. Providing such a comparison could lend even greater specificity and precision to their findings.

Response: We thank Reviewer #1 for the positive opinion of our study and constructive suggestions. Regarding the YAP expression in PNEC, we analyze the single cell RNA Seq data in the Human CellCards Multi-Study CellRef 1.0 Atlas. YAP is expressed in PNEC cells although at a low level but still significantly higher than the 0 level in neutrophils. YAP expression in SCLC is nearly 0, similar to neutrophils (CCLE database). We have added this information in the revised manuscript.

Reviewer #2 (Remarks to the Author):

The revised manuscript from Wu and colleagues is outstanding. The revision is highly responsive to the previous critiques and has now solidified the story in mechanistic detail. The authors have removed some distracting model systems from the previous submission which really improved the clarity of the findings. The data throughout the paper suggest the observations are biologically important and are robust across multiple cell lines. The insights gleaned are highly provocative as they tackle an important issue in SCLC research. The mechanism that links Rb-E2F7-HDAC-YAP to amoeboid cell migration and metastasis is novel and will be of great interest to the SCLC field.

minor points:

Line 155-156: The conclusion is too strong and should be softened. Yes both amoeboid migration and metastasis are inhibited by YAP expression, but these could be unrelated at this point. No experiments separate these phenomena functionally.

Line 290: Typo "F2F7" to "E2F7"

Line 290: Typo "selectively" to "selective"

Response: We thank the reviewer for the high remark of our study and the appreciation of the mechanistic insights revealed during the revision. We have modified the conclusion in lines 155-156 (lines 156-158 in the revised version) to soften the statement. The typo on line 290 regarding "F2F7" has been corrected to "E2F7". However, we were unable to find the "selectively" typo mentioned in line 290.